# Cross-layer transmission realized by light-emitting memristor for constructing ultra-deep neural network with transfer learning ability

Zhenjia Chen [1,2], Zhenyuan Lin[1,2], Ji Yang[3], Cong Chen[1,2], Di Liu[1,2], Liuting Shan[1,2], Yuanyuan Hu [4], Tailiang Guo[1,2] & Huipeng Chen [1,2] ✉

Deep neural networks have revolutionized several domains, including autonomous driving, cancer detection, and drug design, and are the foundation for massive artificial intelligence models. However, hardware neural network reports still mainly focus on shallow networks (2 to 5 layers). Implementing deep neural networks in hardware is challenging due to the layer-by-layer structure, resulting in long training times, signal interference, and low accuracy due to gradient explosion/vanishing. Here, we utilize negative ultraviolet photoconductive light-emitting memristors with intrinsic parallelism and hardware-software co-design to achieve electrical information's optical cross-layer transmission. We propose a hybrid ultra-deep photoelectric neural network and an ultra-deep super-resolution reconstruction neural network using light-emitting memristors and cross-layer block, expanding the networks to 54 and 135 layers, respectively. Further, two networks enable transfer learning, approaching or surpassing software-designed networks in multi-dataset recognition and high-resolution restoration tasks. These proposed strategies show great potential for high-precision multifunctional hardware neural networks and edge artificial intelligence.

Deep neural networks (DNNs) possess the capability to represent more complex nonlinear problems than shallow neural networks, and their distributed data learning method is more effective[1–3]. The development of DNNs has greatly advanced the breakthroughs in autonomous driving[4], cancer detection[5], drug design[6], and they serve as the foundation for massive AI models like ChatGPT[7], PaLM[8], PanguLM[9]. These advancements are largely attributed to the continuous improvement in computational power, network scale, and available data, which enables the implementation of DNNs with more layers and neurons. However, the execution of DNN models demands substantial computational resources. Currently, mainstream methods involve the use of high-end GPUs, accelerators, or cloud computing, which incur high costs and latency and severely limits the application of DNNs in edge AI scenarios, autonomous driving and robotics[10].

Within this background, neuromorphic devices have been intensively studied in recent years, aiming for implementing hardware neural networks with low power consumption and high speed[11–16]. Nonetheless, the current efforts have mainly focused on shallow neural networks with a few layers (typically 2 to 5 layers)[17–21]. Meanwhile,

[1]Institute of Optoelectronic Display, National & Local United Engineering Lab of Flat Panel Display Technology, Fuzhou University, Fuzhou 350002, China. [2]Fujian Science & Technology Innovation Laboratory for Optoelectronic Information of China, Fuzhou 350100, China. [3]College of Computer and Data Science, Fuzhou University, Fuzhou, Fujian, China. [4]Changsha Semiconductor Technology and Application Innovation Research Institute, College of Semiconductors (College of Integrated Circuits), Hunan University, Changsha 410082, China. ✉e-mail: hpchen@fzu.edu.cn

achieving high accuracy in multifunctional DNNs using the reported devices and hardware network structures remains huge challenging. The main reason is that the majority of networks rely on back-propagation (BP) for weight updates[22–25], yet the layer-by-layer structure leads to gradient vanishing (exploding), making it difficult to effectively train the network[26–29]. Although the most advanced neuromorphic chips have attempted to construct DNNs with cross-layer transmission, they still rely on repeated reading of digital memory and DAC/ADC to achieve parallel output of results[30–32]. This greatly limits the throughput of data, as these works construct hardware neural networks through all-electric memristors and traditional software-designed structure instead of designing neural networks based on intrinsic parallel devices. Another issue is that, neuromorphic devices require physical processes for weight modulation[33,34], and more layers and synaptic devices will inevitably increase the training time of DNNs. Therefore, to overcome the limitations of current work in terms of computation and training speed and achieve efficient and versatile hardware DNNs, the key lies in designing innovative fast-modulating intrinsic parallel synapses and building cross-layer modules and neural networks through hardware-software co-design.

Here, a strategy utilizing innovative dual-output N-LEM to construct cross-layer block (ClBlock) was proposed to achieve a hybrid ultra-deep photoelectric neural network (UPENN) and an ultra-deep super-resolution reconstruction neural network (USRNN) with transfer learning ability. The N-LEM reducing the reset time of a single device after 50 enhanced pulses to less than 3.52 % of natural decay time, which can effectively accelerate the training of DNNs. Furthermore, the N-LEM array and hardware-software co-design were employed to achieve the equivalent cross-layer transmission of electrical information using optical signals, which was utilized in the construction of the ClBlock. Based on optical cross-layer transmission strategy, the UPENN and USRNN effectively prevented gradient vanishing (exploding), extended DNN to 54 and 134 layers, and shown strong transfer learning ability. The N-LEM and optical cross-layer transmission strategy successfully filled the gap in the construction of efficient, accurate, high-robust and low-power DNN, providing a new scheme for high-precision multifunctional hardware neural network and edge AI.

## Result

### Schematic diagram of N-LEM and cross-layer transmission structure in ultra-deep neural networks

The complex structure of human brain empowers individuals with strong learning and reasoning abilities, allowing them to expand upon acquired knowledge and engage in logical thinking across diverse fields[35,36]. To achieve a more human-like behavior in artificial neural networks, it is necessary to increase layers and neurons in the networks, thereby enhancing their transfer learning ability. Figure 1a illustrates a schematic diagram of pre-learning method and cross-regional hierarchical structure in biological brain[37–40]. The artificial neural network realized by imitating this structure shows great ability after extensive basic learning (pre-learning) and achieves high accuracy in untrained tasks, which is very similar to human learning ability.

In previous all-electronic hardware neural network implementations, the use of DAC/ADC and memory is necessary for cross-layer signal transmission due to the negative effects of high-frequency modulation and parasitics on signal quality[30]. All-electronic systems typically use ADCs to convert computed synaptic currents into digital signals for storage, and DACs are used to read these digital signals and convert them into corresponding voltage signals in layers requiring access to computed results[17,23]. This process consumes additional computational resources and limits the speed improvement of hardware neural networks as digital signals have lower throughput than analog signals. On the other hand, photonic neural networks

effectively address the issues of high-frequency interference and signal parallelization[10,41]. Optical signals are immune to electrical signal interference, and the propagation of light beams between layers will not cross talk with each other. This has enabled previous reports to achieve end-to-end classification times equivalent to a single clock cycle of state-of-the-art digital platforms[42]. However, photonic neural networks face challenges in being compatible with CMOS technology, and the dense fiber layout occupies a large area. Furthermore, once the hardware network is constructed, changing weights becomes extremely costly as the weights are typically expressed using transmittance, which is fixed and unchangeable. Furthermore, despite some efforts have proposed artificial luminescent synapses, these devices only employed one output (optical or electrical signal) for transmission in the neural network, failing to propose effective strategies to leverage the dual-output potential of devices and achieve powerful networks[13,34,43]. Hence, a strategy utilizing dual-output characteristics of N-LEM to construct cross-layer transmission block (ClBlock) was proposed to achieve an ultra-deep photoelectric neural network (UPENN) and ultra-deep super-resolution reconstruction neural network (USRNN) with transfer learning ability (Fig. 1b), effectively combining the synaptic tunability and integration advantages of hardware electronic neural networks with the interference immunity and parallelization advantages of photonic neural networks. The UPENN and USRNN employ the optical output of N-LEM for ClBlock, effectively avoiding signal interference and leveraging the advantages of one-to-many transmission[34,43], thus alleviating the issue of gradient vanishing (exploding) in deep neural networks.

Figure 1c shows the structure diagram of N-LEM with ultraviolet suppression and dual-output characteristics. The device is capable of generating a photogenerated electric field at the interface of IDTBT/PVP/QDs under UV stimulation, and effectively resetting the synaptic weight. Figure 1d is the schematic diagram of the UV negative photo-conductive effect and parallel equivalent output of the device. The corresponding UV-visible absorption spectrum of the materials is displayed in Figure. S1. Figure. S2 presents the emission spectrum of the device.

### Characteristics of N-LEM with photoelectric modulation and dual output

In Fig. 2a, the output current characteristics of device is observed by three consecutive positive voltage scans (0 V-6 V-0 V) under dark conditions. The presence of the ionic relaxation effect results in a counterclockwise hysteresis in the transfer curve. Additionally, as the number of scans increases, both the conductance and maximum brightness (as shown in the inset of Fig. 2a) continuously increase, which indicates the potential ability of N-LEM as a synaptic device. The voltage-induced synaptic characteristics are primarily associated with hole trapping in the PVP layer, which modifies the device conductance and facilitates hole injection and composite luminescence (Fig. 2b). This process has been previously reported in our work[13], and we have presented a schematic diagram and provided a detailed explanation of this process in Figure. S3. 10, 30 and 50 continuous electric pulses (5 V, 30 ms) were applied to obtain continuous postsynaptic currents, and photodetector and oscilloscope are used to record the post-synaptic brightness of the device (Figure. S4 is the schematic diagram of the test process). Figure 2c illustrates the typical excitatory post-synaptic current (EPSC) and corresponding excitatory postsynaptic brightness (EPSB) generated by 10 voltage pulses. The synaptic device demonstrates excellent synaptic properties, with a strong correlation observed between the light output signal and the electrical signal. The nonlinear (NL) weight updates of the synaptic device were determined by fitting the EPSC and EPSB with the pulses (Fig. 2d), in which yields NL values of 0.20 for electricity and 0.27 for light. The PEARSON correlation coefficient between the two signals reached 0.999, demonstrating the feasibility of using the device for the transmission of the

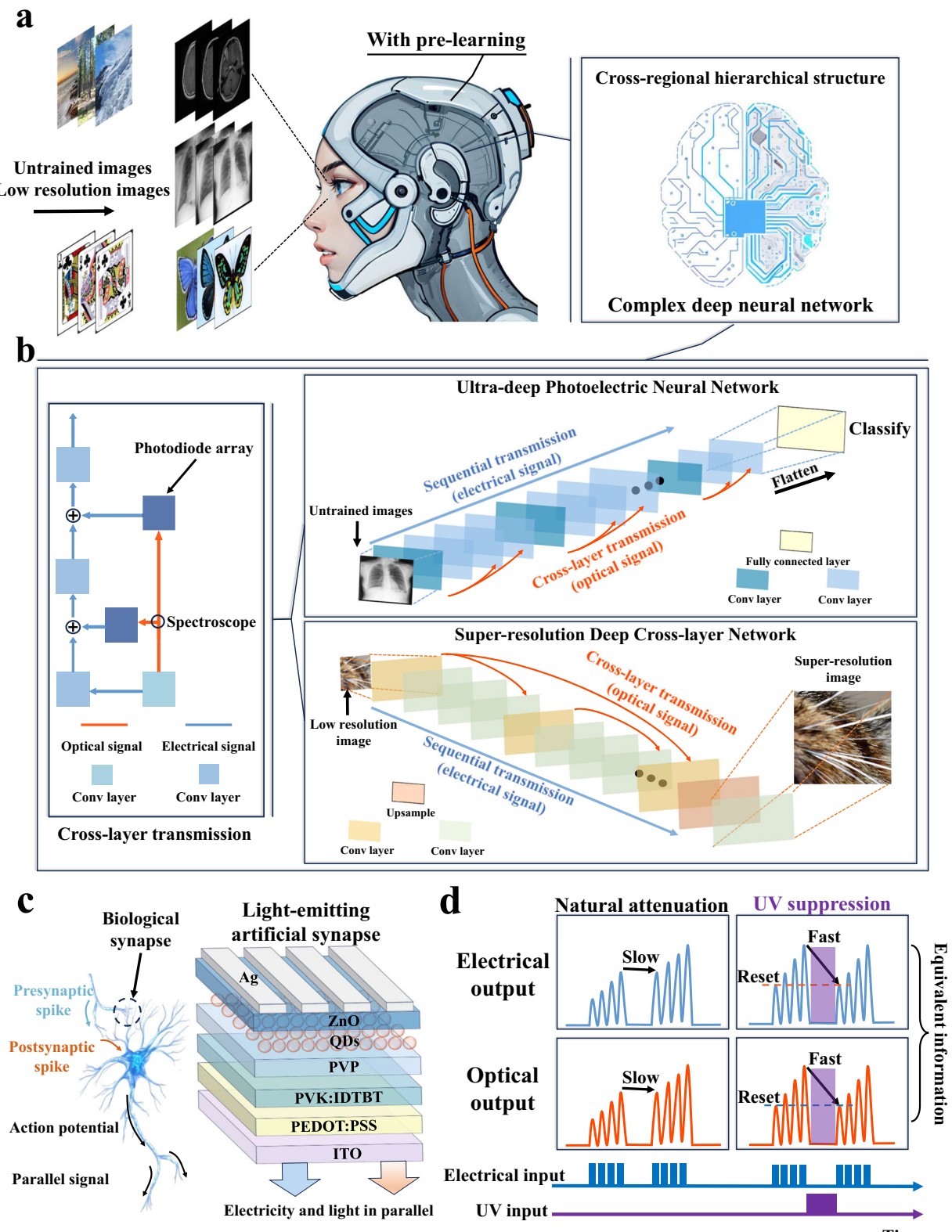

**Fig. 1 | Schematic diagram of UPENN and USRNN with transfer learning ability and the structural diagram of N-LEM with photoelectric equivalent transmission and UV suppression. a** A schematic diagram of pre-learning method and cross-regional hierarchical structure in biological brain. After pre-learning, the brain can process the unlearned content. **b** Schematic diagram of neural network for photoelectric parallel transmission with cross-layer transmission structure. **c** Structure diagram of N-LEM with negative UV photoconductive response and parallel output. **d** Schematic diagram of photoelectric equivalent transmission and UV negative photoconductive phenomenon of N-LEM.

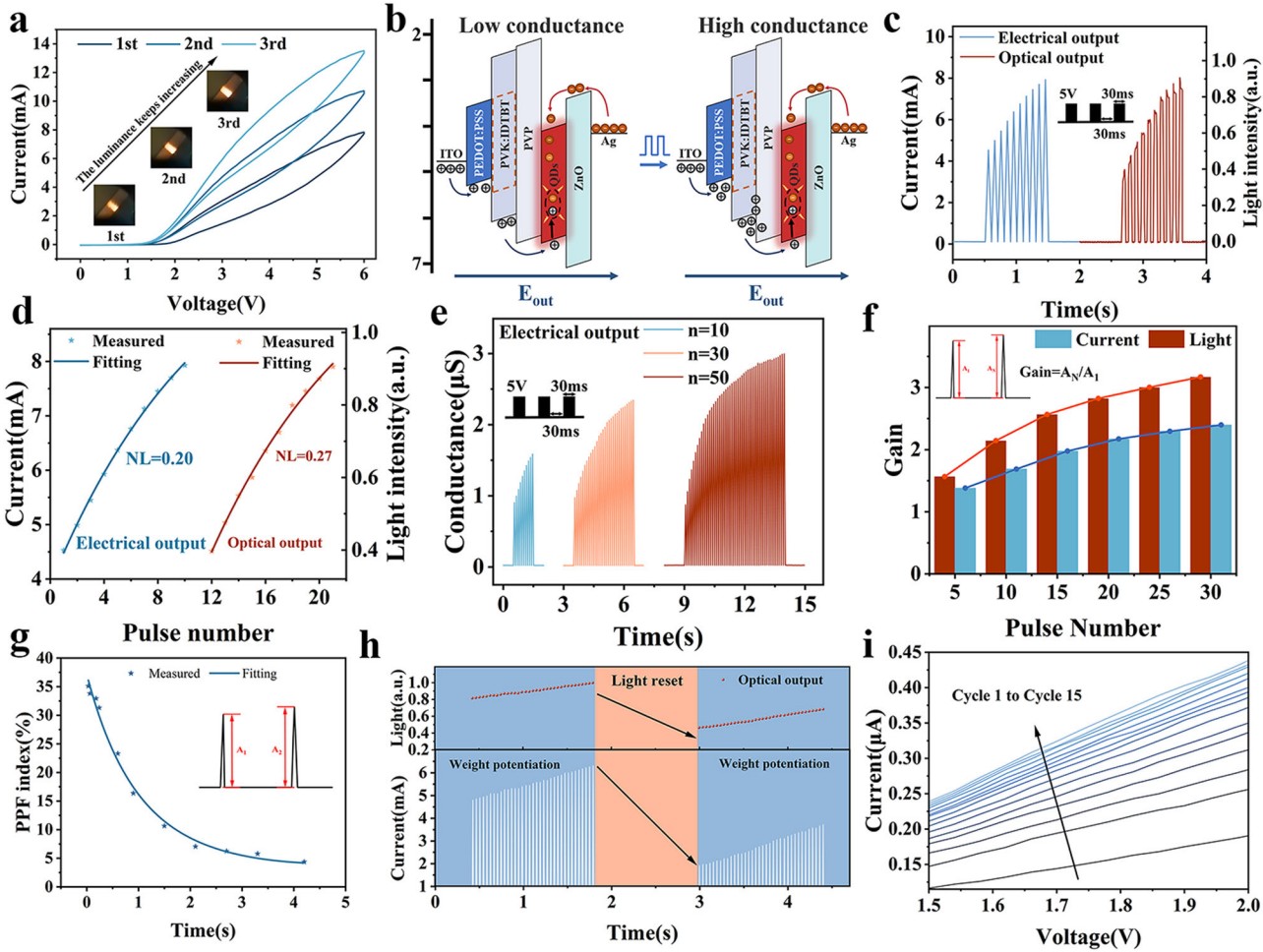

**Fig. 2 | Performance characterization of N-LEM. a** I-V characteristics of N-LEM measured in continuous double-sweep model. **b** Charge trapping process inside the device during electrical pulse stimulation. **c** Current signal and optical signal under 10 continuous voltage pulses. **d** NL fitting of electrical and optical signals of 10 pulses. (**e**) Conductivity changes of devices caused by continuous electric pulses. **f** Gain of EPSC and EPSB under different pulse numbers. **g** Schematic diagram of PPF exponential fitting curve of device. **h** UV irradiation leads to current and light intensity suppression in devices (4 V, 30 ms). **i** Current variation of equipment in 15 consecutive I-V scans.

same information by both light and electricity (Figure. S5). Additionally, the multi-conductance distribution of the device was examined under different pulse conditions ($n = 10$, $n = 30$, and $n = 50$), as shown in Fig. 2e, with the corresponding EPSB displayed in Figure. S6. Moreover, with continuous pulse stimulation, the gain of EPSC (the ratio of the peak value $A_N$ of the last pulse to the first peak value $A_1$) gradually increased from 1.38 to 2.40, while the corresponding gain of EPSB increased from 1.57 to 3.17 (Fig. 2f).

Paired-pulse facilitation (PPF) is a commonly observed phenomenon in short-term synaptic plasticity. To evaluate this effect in our device, two consecutive electrical pulses at varying inter-pulse intervals ($\Delta T$) were applied, and the PPF index was calculated using the formula PPF index = $(A_2 - A_1)/A_1$, as depicted in Fig. 2g. Previous studies reported the breakdown of light-emitting memristors when a large reverse voltage was applied to suppress synaptic weights[34,43]. The N-LEM proposed in this work can realize weight suppression by using photo-generated electric field and avoid reverse breakdown, as shown in Fig. 2h. Figure 2f illustrates the multi-conductance state distribution of our device under 15 consecutive cyclic voltage scans, while Figure. S7 displays the conductance enhancement and suppression curves obtained by applying continuous electrical pulse stimulation and UV stimulation. The device shows excellent distinguishable conductance changes and is suitable for constructing neural networks.

## Mechanism of negative UV photoconductance effect in N-LEM

To investigate the negative UV photoconductivity mechanism of the device, 10 devices are fabricated with different structures (Supplementary Table 1). Figure. S8 displays the current changes of 9 control structures during the process of 50 electrical pulse modulations with or without UV light exposure for 3 s, followed by continuous electrical pulse stimulation, electrical pulses and UV light are applied to all devices under identical conditions. The results reveal that only the device with the IDBT/PVP/QDs(ZnO) exhibited significant negative UV photoconductivity. This effect is attributed to the presence of a built-in electric field ($E_{in}$) generated by photogenerated electrons and holes. As depicted in Fig. 3a, a portion of the photogenerated electrons recombine with the accumulated holes at the interface, leading to a reduction in device conductivity. Moreover, PVP captured a large number of holes (electrons) at the interface of IDTBT(QDs) under the action of electric field, which led to the accumulation of photogenerated electrons (holes). Consequently, a built-in electric field $E_{in}$, opposite to the $E_{out}$, is formed at the interfaces of IDTBT/PVP and PVP/QDs. The magnitude of $E_{in}$ gradually increases with prolonged illumination until it reaches saturation. This phenomenon further hinders the trapping of holes and their migration across PVP into the luminescent layer. To verify this, the physical field coupling of COMSOL Multiphysics was employed to simulate the generation of electric field.

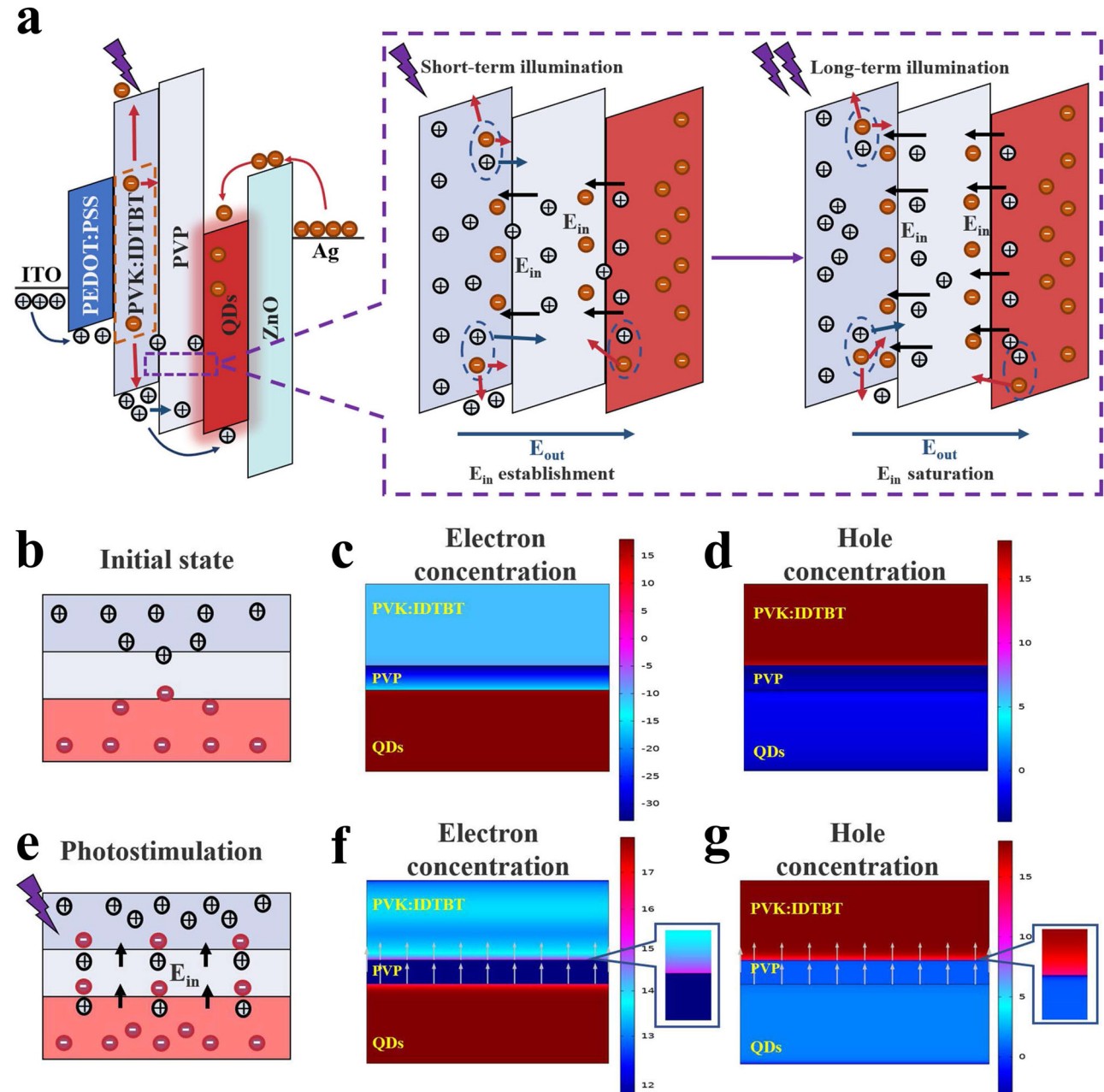

*The scale bars are the result of taking the logarithm with a base of 10 of the carrier concentration, in units of /m³.

**Fig. 3 | Mechanism and simulation of negative UV photoconductance effect.**
**a** Formation and saturation process of built-in electric field in devices caused by UV light. **b** Device model and charge distribution in initial state. **c** Software simulation diagram of electronic distribution of devices in the dark. **d** Software simulation diagram of electronic distribution of devices in the dark. **e** Device model, charge distribution and built-in electric field direction under UV irradiation. **f** Software simulation diagram of electron distribution and electric field direction of the device under UV light. **g** Software simulation diagram of holes distribution and electric field direction of the device under UV light.

Figure 3b and e illustrate the UV suppression principle of the device in the initial state and under illumination, respectively. Figure 3c and d show the distribution of electrons and holes in the initial state, while Fig. 3f and g display the distribution of electrons and holes in the device under UV irradiation. In the initial state, the electron concentration at the interface of PVK:IDTBT/PVP is approximately $10^{-15}$ /m². Under illumination, the photogenerated electrons at the interface are attracted and accumulated by the holes trapped in PVP, resulting in an increase in the electron concentration to around $10^{15}$ /m³, then forming an electric field $E_{in}$ at the interface with the holes in PVP. The presence of $E_{in}$ causes the holes near the PVP layer in IDTBT to repel towards the anode, preventing their movement towards the luminous layer. Similarly, the $E_{in}$ generated at the interface between PVP and QDs can be explained in the same manner, considering that the QDs layer can be considered as N-type material due to the injection of a large number of electrons by ZnO, and PVP can also capture electrons. The arrows at the interface in Fig. 3f and g represent the direction of $E_{in}$ at the interface, as simulated by COMSOL Multiphysics, which is consistent with Fig. 3a. It should be noted that under a large forward voltage, electrons and holes in the QDs layer recombine to emit light, and the $E_{in}$ is mainly produced at the IDTBT/PVP interface. The distribution of the electric field generated by UV illumination from simulation is

shown in Figure. S9. Additionally, under illumination, IDTBT generates a significant number of electron-hole pairs. Since the majority carriers in IDTBT are holes, the hole concentration remains relatively unchanged, while the electron concentration increases. According to the Fermi-Dirac distribution function, the Fermi energy level of materials will rise, resulting in energy band distortion and the increase of potential barrier between IDTBT and PVP, hindering the transmission of holes, as shown in Figure. S10. Furthermore, an organic field effect transistor (OTFTs) is fabricated with the structure shown in Figure. S11 to confirm the generation of an electric field by negative UV photoconductance of the device. And the direction of photo-induced electric field is demonstrated in detail in Figure. S12 to Figure. S14 and Supplementary Note 1. These experiments fully explain the reason behind the negative UV photoconductance of our memristor.

By utilizing the negative UV photoconductance effect, the time required for device reset can be effectively shortened. As depicted in Figure. S15, the first pulse of the second cycle only attenuates by 63.06 % after a natural attenuation of 71.6 s when 50 consecutive electric pulses are applied. However, after 2.52 s of UV irradiation with an intensity of 70 mW/cm$^2$, the weight of the device is reduced by 100 % (Figure. S16) and time cost is reduced to below 3.52 %. Furthermore, in Figures. S17 to S19, we demonstrate the repeatability of this reset operation, the reset time with different pulse settings, and the reset effect under different optical powers within a fixed duration. This offers an efficient method for weight resetting in the training of dual-output memristors in neural networks.

## Cross-layer transmission block based on N-LEM and hardware-software co-design

The dual outputs of N-LEM exhibit strong correlation, which is extremely suitable for realizing cross-layer transmission of equivalent information without other units. A hardware-software co-design approach is used to construct two fully connected networks (FCNs) that share an input layer and a hidden layer, which are utilized to validate the equivalent and cross-layer transmission capabilities of our device for optical signals. The network structure is depicted in Fig. 4a, where the output information from hidden layer 1 needs to be transmitted to hidden layer 2 via synapses. Hidden layer 2 of FCN-1 and FCN-2 respectively receive the post-synaptic current and post-synaptic brightness from hidden layer 1. The detailed circuit structure is illustrated in Fig. 4b. In this case, N-LEMs and photodiodes are designed for weight expression and cross-layer signal reception. Since weight of a single memristor can only generate a positive value, the conductance difference between two memristors is utilized to represent one weight of FCN. FCN-1 includes 50 N-LEMs to represent 25 weights, while FCN-2 is equipped with 50 photodiodes to receive optical signals correspondingly. The current and brightness of each device are measured and adjusted to achieve optimal weight printing. The electrical potential signal generated by the photodiode array is directly proportional to the synaptic current in FCN-1 and is fed into the final layer after analog-to-digital conversion (ADC). All the synapses in the system except between the hidden layers 1 and 2 are implemented in FPGA (represented by the yellow synapse in Fig. 4b). The synaptic connection between hidden layer 1 and hidden layer 2 is established using a customized PCB circuit and N-LEM array. The output of the FPGA is converted into the input signal of the PCB circuit using a digital-to-analog converter (DAC), and the analog signal from the PCB circuit is input into the FPGA through an ADC for final identification.

Limited by the conductance state of the discontinuous hardware synapse device, the neural network designed by software adds weight restriction, which divides the positive and negative values of the weight between the hidden layers 1 and 2 of the neural network into 15 states, accommodating the conductance in the hardware circuit[17]. Furthermore, since FCN has a limited number of neurons in the hidden layer, namely 5, 5, and 20 neurons respectively, this network is first employed for different classification tasks in software to ensure the accuracy of the network. The number of training graphs of MNIST dataset is 5000 and the number of test graphs is 1000. Figure 4c displays the PCB hardware diagram of synaptic connection between hidden layers in FCN-1 and FCN-2, the position of the light-emitting memristor array and the picture of the array are marked. The array used for receiving optical signals is identical to the light-emitting memristor array, except that the devices are replaced with photodiodes. As shown in Fig. 4d, when the classification task is two kinds of classification, the accuracy rate reaches 99 %, and with the increase of classification number, the accuracy rate drops to 73.48 %. We choose the network model with four classifications, write the weights of each epoch into FPGA and N-LEMs (pulses are 1 ms and 5 V). The distribution of 25 weights between hidden layers 1 and 2 after training is shown in Fig. 4e. In the testing phase, FCN-1 achieves a maximum accuracy of 91.6 % while FCN-2 achieves a maximum accuracy of 90.23 %. These values are close to the ideal accuracy of 94.95 % (Fig. 4f). The successful construction of FCN circuit demonstrates the feasibility of utilizing optical signals for cross-layer transmission in deep neural networks. Furthermore, this electro-optical hybrid structure can effectively exert optical advantages, realize low-energy and high-parallel integration technology, and break through the physical limitations of traditional electrical systems[10,34].

## The UPENN based on cross-layer transmission block

The above results demonstrate that the cross-layer block (ClBlock) constructed by N-LEM is reliable. Furthermore, the existing Res-net model has been modified as the ultra-deep photoelectric neural network (UPENN) based on the cross-layer transmission module, with adjustments made to the signal reception layers, as well as the number and positions of cross-layer transmission modules to ensure compatibility with hardware implementation. The UPENN is developed with 53 convolution layers (including 16 cross-layer transmission modules) and a fully connected layer (Fig. 5a and Figure. S20). In addition, a common neural network called Contrast Net is constructed, which does not incorporate cross-layer transmission but has the same other network parameters as UPENN. (Figure. S20 to Figure. S23 show the structures of these two networks in detail).

To elucidate the role of cross-layer transmission in the construction of UPENN, the gradient change in the process of backpropagation of the two networks is discussed in Supplementary Note 2. Moreover, two network models are constructed for the purpose of comparison. The gradient of each layer in the UPENN and Contrast Net is analyzed using supercomputer. The gradient changes from deep to shallow in the first convolution layer of the ClBlocks is emphasized in stage 1 and stage 4 of both networks. (Figure. S21 and S23). The violin diagram in Fig. 5b, derived from the gradient data selected from UPENN, shows changes in the gradient during the internal transmission process of stage 4 and stage 1. Despite the gradient propagating through numerous layers, as illustrated in Fig. 5b, the average gradient μ still exhibits perceptible non-exponential changes, indicating that the shallow layer weights can be updated through training in UPENN. On the contrary, average μ and variance σ of the gradient in Contrast Net decrease by a factor of 10 during the internal transmission process of stage 4 (Fig. 5c), and after the propagation of multiple layers, the average μ and variance σ of the gradient in stage 1 converge to approximately $10^{-8}$. Consequently, the gradient of the deep layer diminishes significantly when it is propagated to shallow layers, resulting in ineffective training and low accuracy. The above analysis demonstrates the effectiveness of cross-layer optical transmission in addressing the issue of gradient vanishing (exploding) in DNNs.

## Transfer learning and multi-task recognition based on UPENN

Different from shallow networks that can only perform single task, the ultra-deep photoelectric neural network (UPENN) is equipped with

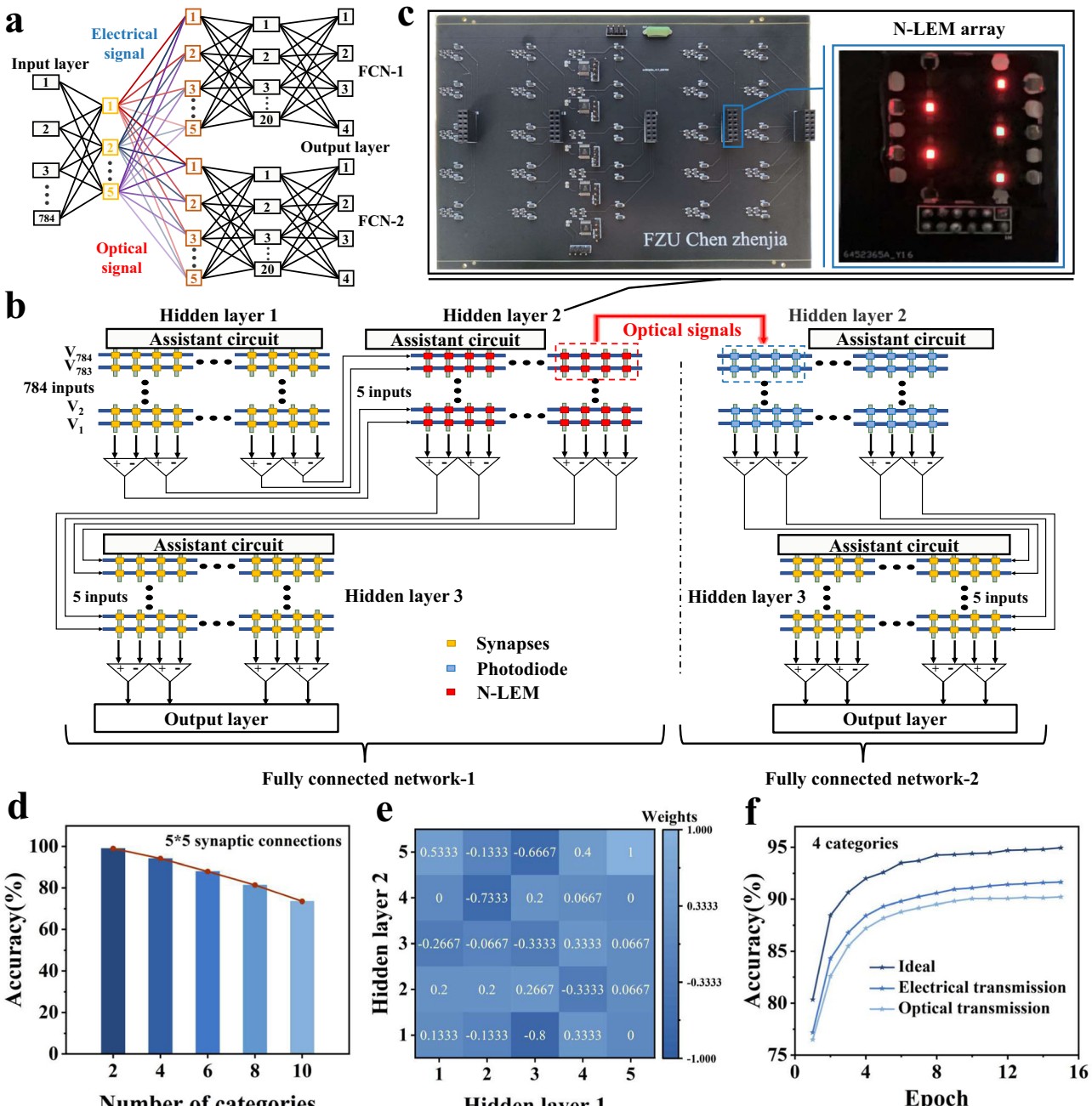

**Fig. 4 | Optical cross-layer transmission of equivalent electrical signal by hardware-software co-design. a** Schematic diagram of two neural networks sharing input layer and hidden layer 1, using optical signals and electrical signals to transmit equivalent information. **b** Circuit diagram of FCN-1 and FCN-2. **c** Hardware circuit and N-LEM array for realizing synaptic connection between hidden layers 1 and 2. **d** Schematic diagram of network accuracy decreasing with the increase of categories. **e** Inter-layer weight distribution (between hidden layers 1 and 2) obtained from the training of four digital classification tasks. **f** The accuracy of FCN-1 and FCN-2 obtained by writing the weight of each epoch into FPGA and circuit changes.

substantial neurons and synapses, and the cross-layer transmission modules enable it to be effectively trained, thus possessing transfer learning ability similar to the human brain. To demonstrate this advantage, a large number of datasets from ImageNet are used to pre-learn UPENN and Contrast Net. This enables them to effectively extract and recognize image features, similar to the innate capability of the human brain. Subsequently, the accuracy of the networks is evaluated using untrained 8 datasets. The process of pre-learning and re-learning is depicted in Fig. 5a. The number of types and recognition accuracy of each dataset can be found in Supplementary Table 2, and hyperlinks are provided to access the datasets. This validation of transfer learning

ability is conducted by directly using the test sets of 8 datasets without any additional training.

Figures 5d to 5f depict the feature maps obtained from two neural networks using X-ray chest images of COVID-19 patients for pneumonia diagnosis. The original images labeled as negative retain rich features after passing through different convolutional kernels in UPENN network during the propagation from shallow to deep layers (from stage 1 to stage 4), while the feature maps in Contrast Net tend to be consistent after convolution (Contrast Net fails to obtain effective convolutional kernels due to gradient vanishing). To demonstrate the feature extraction capabilities of the two networks more intuitively,

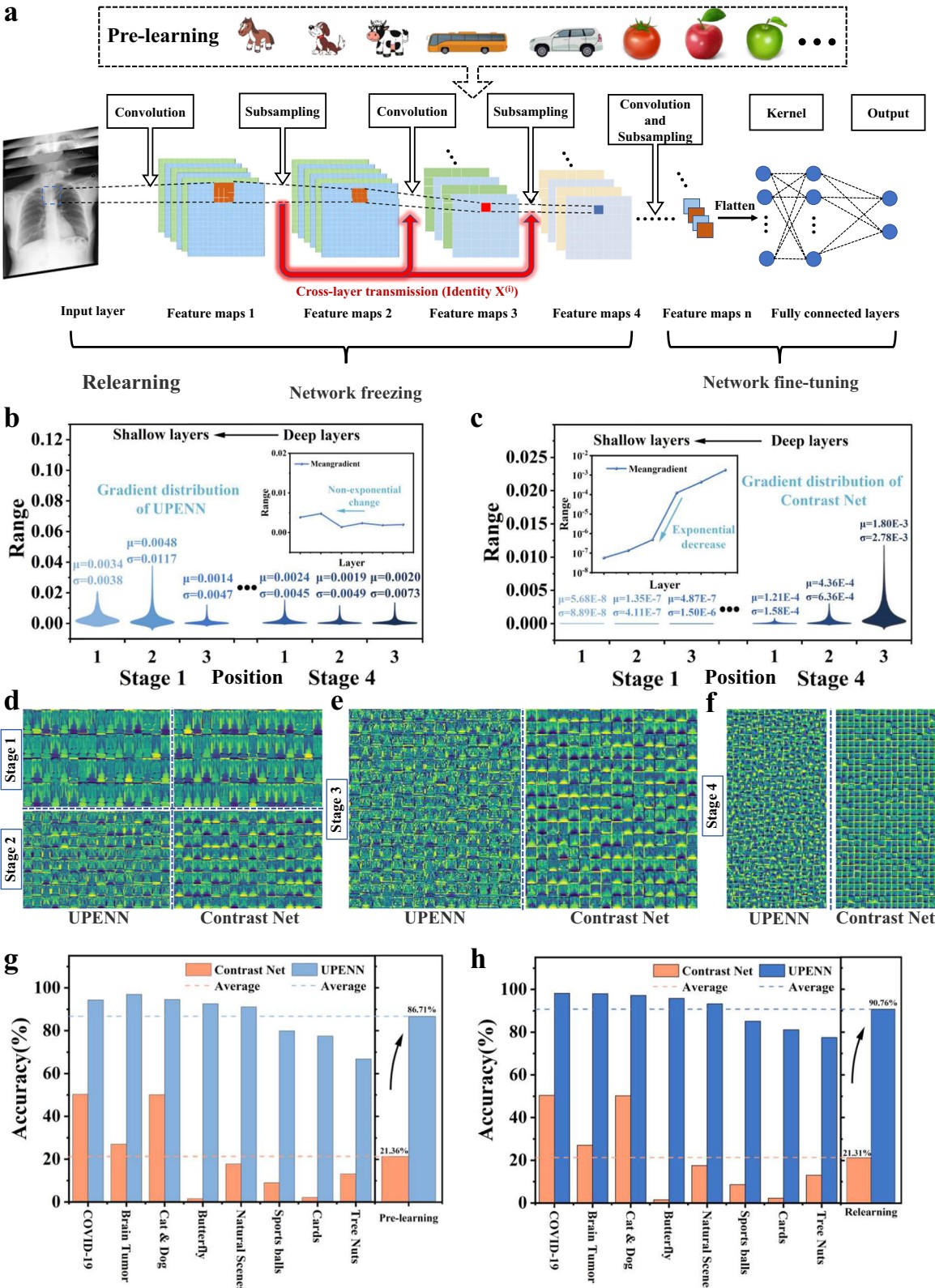

**Fig. 5 | Structure and performance of UPENN and Contrast Net. a** Schematic diagram of UPENN and cross-layer transmission, marked with pre-learning and re-learning methods. **b**, **c** Gradient distribution calculated by selecting specific layers of stage 1 and stage 4 in UPENN and Contrast Net. **d** Feature maps extracted from the same photo in stage 1 and stage 2 of two networks. **e** The feature maps extracted from the same photo in stage 3 of two networks, the content of Contrast Net has lost most of its features. **f** Characteristic Diagram in stage 4 of Two Networks. **g** The recognition accuracy of UPENN and Contrast Net on the unlearned content after pre-learning. **h** The recognition accuracy of UPENN and Contrast Net after re-learning.

two X-ray images labeled as negative and positive are input into the two networks, and the feature maps are obtained at the same network positions (Figure S24 and S25). The similarity of the feature maps obtained from the same network for both images is analyzed using the structural similarity index (SSIM) to determine whether the network can effectively distinguish between negative and positive. The SSIM for UPENN and Contrast Net are 0.052 and 0.925, respectively (closer to 1 indicates higher similarity), indicating that the features obtained after convolution in UPENN are extremely different, which helps UPENN to effectively distinguish between negative and positive cases. On the other hand, the feature maps obtained by Contrast Net are highly similar, suggesting that this network is highly likely to classify the two conditions into the same category. Therefore, Contrast Net has a very low accuracy (close to the minimum value of 50 % for binary classification tasks). The accuracy and average performance of both networks in the pre-learning phase are shown in Fig. 5g. The Contrast Net exhibits a decrease in accuracy as the types in dataset increases, with an average accuracy of 21.36 %. In contrast, the UPENN achieves an average accuracy of 86.71 %, which is 305.9 % higher than the Contrast Net.

In addition, the relearning method is used to improve the accuracy of UPENN in specific tasks, and to verify whether the Contrast Net can achieve higher accuracy on specific tasks after targeted training. During the re-learning, most of the neural networks are frozen, allowing only adjustments to the weights of the last fully connected layer and a ClBlock (as shown in Fig. 5a), while the Contrast Net is allowed to adjust the whole network to achieve higher accuracy. After relearning, the Contrast Net struggles due to gradient vanishing, resulting in an average accuracy of only 21.31 %. Conversely, UPENN achieves an average accuracy of 90.76 %, which is 325.9 % higher than the Contrast Net (Fig. 5h). Therefore, relearning can effectively improve the accuracy of UPENN for specific tasks and demonstrate its practicability. These results confirm the transfer learning ability of UPENN based on cross-layer transmission and N-LEM. Additionally, UPENN has not yet reached its layer limit, indicating the potential to further increase the number of layers and enhance its functionality.

## The USRNN based on cross-layer transmission block

During the process of image acquisition, most obtained images suffer from suboptimal resolution due to factors such as sensor unit density and optical blurring[44,45]. Consequently, there is an exigent requirement for a technology capable of augmenting spatial resolution of images within the confines of existing hardware capabilities. Super-resolution restoration entails the generation of high-resolution images from blurred low-resolution counterparts[46,47]. Presently, this technology relies on software-based signal processing methods and necessitates high-performance GPUs for super-resolution processing, while being constrained by the throughput and power limitations inherent in traditional architectures.

Here, we propose an ultra-deep super-resolution reconstruction neural network (USRNN) constructed using cross-layer block (ClBlock), thus affirming the versatility of the ClBlock module in building neural networks with diverse functionalities, and furnishing a viable hardware implementation for high-speed, low-power super-resolution processing. In contrast to UPENN, our design features an alternative structure for the ClBlock module, as illustrated in Fig. 6a, with the constructed USRNN comprising 133 layers for 2x resolution restoration and 135 layers for 4x resolution restoration. The detailed ClBlock and USRNN structures are shown in Figure. S26. Following training of the USRNN for 2x resolution restoration, the majority of the USRNN is immobilized, replacing the Upsample module, and allow only fine-tuning of the Upsample and final convolution layers to achieve 4x restoration.

The RGB input blocks of low-resolution (LR) images, sized 48*48, and their corresponding high-resolution (HR) images from the DIV2K dataset are utilized to showcase the restorative capabilities of the USRNN[48]. Furthermore, the hardware-based USRNN, structured with

ClBlock, is compared against existing software-constructed networks, including VDSR[49], SRCNN[50], and SRresnet[51]. A comparison of the peak signal-to-noise ratio (PSNR) and structural similarity index (SSIM) of the Y-channel of multiple models reveals the 2x resolution restoration effects, as depicted in Fig. 6b. The "bicubic" refers to the image that has been enlarged through interpolation.

In contrast to alternative methods, the hardware architecture of USRNN successfully reconstructed details and edges in the high-resolution (HR) image, surpassing the existing software-based approaches in both PSNR and SSIM, achieving performances of 28.94 dB and 0.8975, respectively. Moreover, the fine-tuned 4x resolution restoration USRNN demonstrates robust performance, as evidenced in Fig. 6c and Fig. 6d, which showcases the effects of 2x and 4x resolution restoration on the same image. This highlights the exceptional pre-learning capabilities of the USRNN based on the ClBlock architecture, and its potential for further expansion and application in other super-resolution domains such as X-ray imaging, angiography analysis, and astronomical observations. Figure 6e displays the comparison of PSNR and SSIM obtained by various methods for the images shown in Fig. 6b, while Fig. 6f presents the PSNR performance of USRNN during 2x and 4x training cycles. Additionally, the magnified images of Fig. 6b, c, and d are further depicted in magnified form in Figures. S27–29, allowing for examination of the resolution reconstruction effects and details.

Finally, the advantages of ultra-deep neural networks based on ClBlock are summarized in the following aspects: (i) Efficient training: The device can be reset using UV light, eliminating the reverse breakdown issue and reducing the reset time of 50 enhance pulses to less than 3.52 %. (ii) High accuracy cross-layer transmission: The ClBlock with dual signals is proposed in our devices. Through hardware-software co-design, the optical cross-layer transmission of electrical information is successfully verified, achieving recognition accuracies of 91.66 %(electricity) and 90.23 %(light). (iii) Deeper neural networks: The proposed UPENN and USRNN demonstrate effective prevention of gradient vanishing in the network, enabling the expansion of hardware neural networks from the traditional few layers (typically 2 to 5 layers) to 54 layers and 135 layers. (iv) Excellent transfer learning ability: Compared to networks without cross-layer transmission, UPENN exhibits basic recognition ability for unknown tasks after pre-learning and achieved accuracies of 86.71 % and 90.76 % after pre-learning and relearning, surpassing traditional methods by 305.9 % and 325 % respectively. (v) Software-level performance: The ClBlock-based USRNN demonstrates software-level performance in super-resolution restoration, approaching or surpassing the performance of current software-designed super-resolution restoration networks, achieving a PSNR of 35.15 dB in 2x resolution reconstruction.

## Discussion

In summary, an innovative dual-output N-LEM with negative photoconductivity for ultraviolet light has been developed, significantly reducing the time required for weight suppression. Through software-hardware co-design, the cross-layer transfer module (ClBlock) of neural networks has been successfully implemented. Furthermore, based on ClBlock, two ultra-deep neural networks for multi-task classification and super-resolution restoration have been constructed, effectively expanding the depth and alleviating the issue of gradient vanishing (exploding). The proposed neural networks demonstrate strong transfer learning capabilities and practicality, with UPENN achieving 305.9 % and 325.9 % higher accuracy than Contrast Net after pre-learning and relearning, respectively, while USRNN's performance in resolution restoration approaches or even surpasses that of software-designed networks. Thus, the proposed N-LEM and ClBlock provide solutions for the challenges of training speed and network depth in neural networks, offering new possibilities for hardware-based ultra-deep neural networks and edge AI.

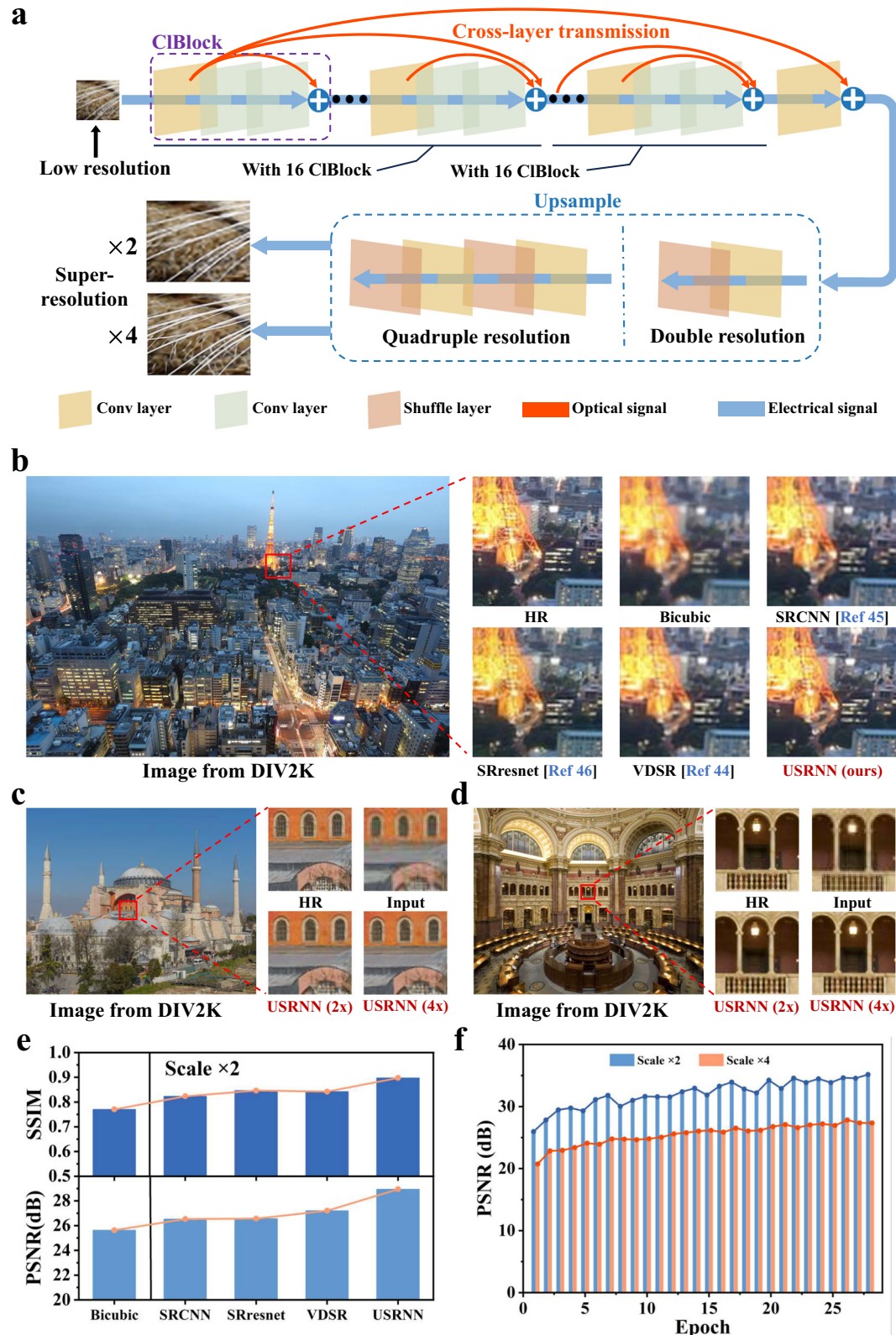

**Fig. 6 | Structure and performance of USRNN. a** The network structure diagram of the USRNN used for super-resolution restoration, capable of achieving 2x or 4x resolution reconstruction. **b** The comparison of the 2x resolution reconstruction effects between the USRNN based on hardware-software co-design and other software methods within the same image. **c** and **d** The demonstration of the effectiveness of the USRNN for 2x and 4x resolution reconstruction. **e** The PSNR and SSIM indices obtained for each image in Fig. 6b. **f** The performance of the USRNN for 2x and 4x resolution reconstruction across multiple training cycles.

## Methods

### Materials availability

The indium tin oxide (ITO) substrates were obtained from Shenzhen Huanan Xiangcheng Technology Corp. The red CdSe/ZnS QDs were obtained from Poly OptoElectronics Co. Ltd. Dissolve Indacenodithiophene–benzothiadiazole (IDTBT) and Poly(9-vinylcarbazole) (PVK) in chlorobenzene at the concentrations of 5 mg/mL and 8 mg/mL respectively, and mix them according to the volume of 1:1. ZnO nanoparticles (ZnO NPs) were synthesized by a solution method. The cross-linked PVP solution was prepared by mixing 150 mg of Poly(4-vinylphenol) (PVP) powder with 15 mg of 4,4′-(hexa-fluoroisopropylidene)-diphthalic anhydride in 1 ml of Propylene glycol monomethyl ether acetate solvent.

### Device fabrication

The PEDOT:PSS solution was spin-coated on the ITO substrate after plasma treatment at a speed of 4000 rpm for 40 s, and then annealed at 120 °C for 10 min. The PVK:IDTBT solution was spin-coated on PEDOT:PSS film at 3000 rpm for 40 s, and then annealed at 120 °C for 10 min. The cross-linked PVP solution was spin-coated on PVK:IDTBT film in nitrogen atmosphere, rotated at 1000 rpm for 5 s and 2000 rpm for 30 s, then annealed at 120 °C for 2 h. The quantum dot solution was spin-coated at 3000 rpm for 40 s, and then annealed at 60 °C for 10 min. The ZnO solution was spin-coated on the luminescent layer at the speed of 3000 rpm for 40 s, and then annealed at 120 °C for 30 min. Finally, 50 nm silver cathode was deposited by vacuum mask to obtain the devices.

### Memristor array fabrication

The patterned ITO glass substrate was customized, so that the ITO electrodes of two adjacent devices were staggered. After the same steps in the device fabrication, the materials between two adjacent ITO electrodes were etched away. Then, the substrate was placed in a patterned mask, and 50 nm Ag electrode was deposited in vacuum. In the obtained device array, the anode ITO of the device was in contact with the anode Ag of the next device. After the array was packaged, the electrodes were led out through the adapter for testing the hardware circuit. And according to the position of each light-emitting memristor, the same photodiode circuit was designed on another PCB circuit for receiving optical signals.

### Device characterization

Transient EL measurements of the QLED were detected using a photodetector (EOT Silicon PIN Detector ET-2030) and an oscilloscope (Keysight DSOX 1202). The brightness and EL spectrum were calculated from a Src-200 spectral color luminometer. The absorption spectra of each layer were obtained by using a spectrophotometer (Shimadzu UV 3600). The electrical characteristics of the synaptic device were measured using Keithley B2902A. Using FPGA to write and test PCB circuits.

## Data availability

Source data are provided with this paper. The data that support the plots within these paper and other findings of this study are available from the corresponding authors upon request.

## Code availability

The code that supports the theoretical plots within this paper is available from the corresponding author upon request.

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

## Acknowledgements

The authors are grateful for financial support from National Natural Science Foundation of China (U21A20497, 62374033), and Fujian Science & Technology Innovation Laboratory for Optoelectronic Information of China (2021ZZ129).

## Author contributions

H.P.C. and Z.J.C. conceived the project. Z.J.C. was responsible for device fabrication, data acquisition and analysis, physical field simulation calculations, circuit design and writing. J.Y. and Z.Y.L. were responsible for the conception and analysis of device applications. C.C., D.L. and L.T.S. were responsible for network guidance and mechanism analysis. Y.Y.H. and T.L.G. were responsible for overseeing the overall work. H.P.C. supervised the project. Everyone has read the manuscript.

## Competing interests

The authors declare no competing interests.
