## [Peer Review File · Nature Communications]

REVIEWER COMMENTS

Reviewer #1 (Remarks to the Author):

In this manuscript, the authors propose a hardware strategy utilizing a light-emitting memristors (N-LEMs) with negative ultraviolet photoconductivity to achieve rapid weight modulation and cross-layer transmission block without DAC/ADC. This N-LEM not only possesses intrinsic parallelism in optoelectronics but also significantly reduces the weight reset time of the device. Furthermore, the concept of leveraging the strong correlation between luminescent memristors and electrical signals for cross-layer communication undoubtedly leaves a deep impression. The construction of two highly deep neural networks using this cross-layer transmission structure has yielded exceptional results in terms of layer depth and performance, which is highly gratifying. However, before publication in Nature Communications, the manuscript still requires some minor revisions. The authors should provide explanations and improvements in the following aspects:

1. Is it feasible to use common electrical synapses to construct cross-layer structures? What are the advantages of N-LEM for the construction of such cross-layer structures?
2. Does the synapses labeled in Figure. 4b require the use of your proposed N-LEM device? What synapse characteristics are required for the complete hardware implementation of this network? Can the majority of artificial synapses be combined with the N-LEM proposed by your team to construct this structure?
3. The Figures shown in Figure. 5d-f depict the intermediate feature maps of the same image in two different networks. The author states that during the convolution process of the two networks, the features in the Contrast Net feature map gradually disappear, resulting in a very low final recognition accuracy. However, it is somewhat challenging to comprehend from this image alone. The reviewers hope that the author can provide additional images to aid in the explanation.
4. The cross-layer module in UPENN is referred to as BTNK, which differs from the CIBlock structure in USRNN. It is necessary for the authors to uniformly name and explain this module in UPENN to facilitate reader comprehension.

Reviewer #3 (Remarks to the Author):

The authors investigate a hybrid ultra-deep photoelectric neural network (UPENN) and an ultra-deep super-resolution reconstruction neural network (USRNN) with transfer learning ability with multiple cross-layer block (CIBlock) by utilizing dual output light-emitting memristors (N-LEMs) with negative ultraviolet photoconductivity. The current efforts have mainly focused on shallow neural networks with a

few layers (typically 2 to 5 layers). The challenges of back propagation for weight update, gradient vanishing, requirement of repeated reading of digital memory, long training time in the current deep neural network (DNN) were addressed in this manuscript. The authors succeed in the construction of efficient training, high accuracy cross layer transmission, excellent transfer learning ability and software-level performance in super-resolution restoration in ultra deep neural network with 54 and 135 layers. The authors should address the following questions and comments.

Comments:

1. The authors state in the introduction that deep neural networks (DNNs) with cross-layer transmission rely on repeated reading of digital memory and require physical processes for weight modulation. Also, DNNs with more layers and synaptic devices increase its training time. These limitations can be overcome by designing innovative fast-modulating intrinsic parallel synapses and building cross-layer modules for direct analog signals transmission in neural networks as mentioned by the authors. The authors should discuss in detail with literature reports on the innovation to overcome the limitations in this part of the introduction, e.g. what are the limitations and innovations?
2. The current-voltage characteristics in the N-LEM were investigated only in the positive voltage range. It is advised to investigate also in the negative voltage range.
3. The voltage induced synaptic characteristics were analyzed as hole trapping in PVP layer. The authors need to explain this.
4. As analyzed by the authors, after 2.52 s of UV irradiation of 50 pulse stimulation with an intensity of 70 mW/cm², the weight of the device is reduced by 100%. The authors should check the repeatability of the resetting time. They should show the consistency of resetting time with some number of setting and resetting operation. The also should show the resetting time with UV intensity.
5. In supplementary Table-1, the UV negative photoconductive characteristics of various structures are indicated. As mention in the text, only the device with the IDBT/PVP/QDs(ZnO) exhibited significant negative UV photoconductivity. The authors need to put the exact photoconductivity value for different structures.
6. The Fig. 3(c, d and f, g) are not clear. Are these figures only show electron distribution or hole distribution? If so, what is the meaning of positive and negative value according to the scale-bar. Also, it is recommended to keep same scale-bar with unit for all the figures.
7. Fig. S5, 10, 12, 13. show actual experimental data of the devices. The authors should not use 'schematic diagram...' in the caption.
8. In many places there is no gap between numeric value and unit. The authors need to correct in whole manuscript.
9. In the insets of Fig. 5(b and c) there is no label on x-axis.
10. It is recommended to use less abbreviations. Especially for a none expert reader, it is difficult to follow e.g. CLBlock, UPENN,USRNN. If possible, please avoid. Also, while I appreciate the extensive

supplementary material, the reference to the supplementary material is too frequent. Maybe, the authors find a way to limit the reference to the supplementary material.

11. While Figure 1 (a) to (c) look visually appealing, the content of the figures is questionable. Especially, Figure 1(a) does not provide the learning ability using a cross regional hierarchical structure.

Response to the reviewers' comments

We thank all the reviewers for their insightful comments on our manuscript. Reviewer's comments are in blue, while our responses are immediately below each comment. We have added relevant experimental results to clarify the concerns of reviewers. The content of the article is optimized, and the modifications to the manuscript are highlighted in the manuscript itself.

Reviewer #1

Comments:

In this manuscript, the authors propose a hardware strategy utilizing a light-emitting memristors (N-LEMs) with negative ultraviolet photoconductivity to achieve rapid weight modulation and cross-layer transmission block without DAC/ADC. This N-LEM not only possesses intrinsic parallelism in optoelectronics but also significantly reduces the weight reset time of the device. Furthermore, the concept of leveraging the strong correlation between luminescent memristors and electrical signals for cross-layer communication undoubtedly leaves a deep impression. The construction of two highly deep neural networks using this cross-layer transmission structure has yielded exceptional results in terms of layer depth and performance, which is highly gratifying. However, before publication in Nature Communications, the manuscript still requires some minor revisions. The authors should provide explanations and improvements in the following aspects.

Response: Thank you very much for your valuable instructions and comments for improving our manuscript. We have made the following supplements and explanations for your questions and suggestions:

1. Is it feasible to use common electrical synapses to construct cross-layer structures? What are the advantages of N-LEM for the construction of such cross-layer structures?

Response: We gratefully appreciate for reviewer's valuable suggestions. At present, most of the work of constructing neural networks with cross-layer transmission is based on electrical memristors, so it is feasible to construct cross-layer structures using electrical synapses. However, the electrical memristor produces a single output, it is extremely difficult to directly divide the output current into multiple outputs of the same size and enter the different layers (because the synaptic weights in a neural network are almost different). As we emphasized in the introduction, "... they still rely on repeated reading of digital memory and DAC/ADC to achieve parallel output of results.", At present, the parallel output of results can only be realized by digital memory and DAC/ADC. For N-LEM, the light output intensity has a strong correlation with the current output, these two analog signals can be regarded as carrying the same information, and because of the easy parallelism and anti-electromagnetic interference characteristics of optical signals, they are very suitable for constructing cross-layer transmission structures. There is no need to use memory and DAC/ADC to read and output current to different neural network layers for many times, and the interference of analog signals in circuit transmission is effectively avoided when optical signals are used across layers.

2. Does the synapses labeled in Figure. 4b require the use of your proposed N-LEM device? What synapse characteristics are required for the complete hardware implementation of this network? Can the majority of artificial synapses be combined with the N-LEM proposed by your team to construct this structure?

Response: We gratefully appreciate for reviewer's valuable suggestions. The synapse labeled in Figure 4b does not necessarily need to use our proposed N-LEM, which is why we use different colors to represent it with N-LEM. Because N-LEM belongs to light-emitting devices, and the power consumption of light-emitting devices is usually greater than that of electrical synapses, in order to realize direct cross-layer transmission and reduce power consumption as much as possible, it is a more ideal scheme to use electrical memristors as the calculation unit of neural networks without cross-layer, so it is only necessary to use common electrical synapses with multi-level conductivity in this part. Most artificial synapses can theoretically combine with our N-LEM to construct direct cross-layer neural networks as long as the conductivity is maintained and distinguished.

3. The Figures shown in Figure. 5d-f depict the intermediate feature maps of the same image in two different networks. The author states that during the convolution process of the two networks, the features in the Contrast Net feature map gradually disappear, resulting in a very low final recognition accuracy. However, it is somewhat challenging to comprehend from this image alone. The reviewers hope that the author can provide additional images to aid in the explanation.

Response: We gratefully appreciate for reviewer's valuable suggestions. We agree with the reviewer that it is difficult to describe the process of feature disappearance only by using a negative COVID-19 diagnosis result as the feature map of the input image in two networks, so we supplement the feature map of another image with a positive diagnosis result in two neural networks as a comparison. And through the structural similarity index (SSIM), we can get the similarity of the final output feature maps of the two images in UPENN and Contrast Net (for 2 classification tasks, the lower the similarity, the higher the classification accuracy). The final result shows that the SSIM of the feature maps of the two input images in UPENN is 0.052 (the closer the SSIM is to 1, the more similar it is, and 0 means completely different), and the SSIM in Contrast Net is 0.925, which indicates that the final features of the two input images in Contrast Net are extremely similar, because their features are lost in the continuous invalid convolution, while completely different features are extracted in UPENN. The original images and feature maps in two neural networks are shown in **Figure. S19** and **Figure. S20**, and we have modified original manuscript, and discussed it in detail under the supplementary images in SI.

For this problem, we have added corresponding supplements in the article and SI, as follows:

In manuscript:

“Figures. 5d to 5f depict the feature maps obtained from ... has a very low accuracy (close to the minimum value of 50% for binary classification tasks).”

In SI:

The previous **Figure. S19** to **S22** were removed, and new **Figures. S24** and **S25** were added, as shown below:

Supplementary Figure. S24 The feature maps of two X-ray images in UPENN and the SSIM of two feature maps in stage 4.

Supplementary Figure. S25 The feature maps of two X-ray images in Contrast Net and the SSIM of two feature maps in stage 4.

“**Figure. S24** shows the feature maps obtained from stage 1 to stage 4 in UPENN for the X-ray images labeled as negative and positive in a binary classification task. In an ideal scenario, the neural network should extract completely different features for negative and positive cases in order to achieve close to 100% accuracy in distinguishing all images into two categories. It is evident from the figure that the two images acquire different features under the influence of the same convolutional kernels (the convolutional kernels are consistent at each position of the feature map), particularly in stage 3 and stage 4. This indicates that UPENN can effectively extract features by successfully avoiding gradient vanishing through cross-layer propagation and obtaining effective convolutional kernels through pre-training. By using SSIM to analyze the similarity between the two feature maps in stage 4, the SSIM coefficient is only 0.052, indicating that the two maps are extremely dissimilar. Therefore, UPENN is capable of effectively

distinguishing between negative and positive cases.”

“**Figure. S25** shows the feature maps obtained from stage 1 to stage 4 in Contrast Net for the X-ray images labeled as negative and positive in a binary classification task. In an ideal scenario, the neural network should extract completely different features for negative and positive cases to achieve close to 100% accuracy in distinguishing all images into two categories. It is obvious from the figure that the two images have obtained very similar features under the influence of the same convolution kernel. This indicates that the convolution kernel of Contrast Net cannot effectively distinguish the image features of the two cases, which is essentially due to the invalid convolution kernel obtained in the pre-learning process caused by the gradient vanishing. By using SSIM to analyze the similarity between the two characteristic graphs in Stage 4, the SSIM coefficient is 0.925, which indicates that the two graphs are very similar. Therefore, Contrast Net cannot effectively distinguish negative and positive cases.”

4. The cross-layer module in UPENN is referred to as BTNK, which differs from the CIBlock structure in USRNN. It is necessary for the authors to uniformly name and explain this module in UPENN to facilitate reader comprehension

Response: We gratefully appreciate for reviewer’s valuable suggestions. We agree with the reviewer's suggestion to provide an explanation and unify BTNK and CIBlock. In fact, BTNK 1 and BTNK 2 are two parts of CIBlock, where BTNK 1 serves as the starting point for optical signal cross-layer transmission, and BTNK 2 acts as the receiving end for optical signal cross-layer transmission. To avoid confusion or misconceptions between CIBlock and BTNK, we have made consistent modifications throughout the main text, replacing BTNK-related content with CIBlock. Additionally, we have provided supplementary explanations below the network architecture diagram of UPENN in the Supplementary Information (SI).

For this problem, we have added corresponding supplements in the article and SI, as follows:

In manuscript:

“The gradient changes from deep to shallow in the first convolution layer of the CIBlocks is emphasized in stage 1 and stage 4 of both networks. (**Figure. S21** and **Figure. S23**).”

In SI:

“Stage 1 to Stage 4 are different structures of CIBlock. CIBlock consists of BTNK 1 and BTNK 2, where BTNK 1 is the transmitting end of the optical interlayer signal, and BTNK 2 is the receiving end of the optical interlayer signal.”

Reviewer #3

Comments:

The authors investigate a hybrid ultra-deep photoelectric neural network (UPENN) and an ultra-deep super-resolution reconstruction neural network (USRNN) with transfer learning ability with multiple cross-layer block (CIBlock) by utilizing dual output light-emitting memristors (N-LEMs) with negative ultraviolet photoconductivity. The current efforts have mainly focused on shallow neural networks with a few layers (typically 2 to 5 layers). The challenges of back propagation for weight update, gradient vanishing, requirement of repeated reading of digital memory, long training time in the current deep neural network (DNN) were addressed in this manuscript. The authors succeed in the construction of efficient training, high accuracy cross layer transmission, excellent transfer learning ability and software-level performance in super-resolution restoration in ultra deep neural network with 54 and 135 layers. The authors should address the following questions and comments.

Response: Thank you very much for your valuable instructions and comments for improving our manuscript. We have made the following supplements and explanations for your questions and suggestions:

1. The authors state in the introduction that deep neural networks (DNNs) with cross-layer transmission rely on repeated reading of digital memory and require physical processes for weight modulation. Also, DNNs with more layers and synaptic devices increase its training time. These limitations can be overcome by designing innovative fast-modulating intrinsic parallel synapses and building cross-layer modules for direct analog signals transmission in neural networks as mentioned by the authors. The authors should discuss in detail with literature reports on the innovation to overcome the limitations in this part of the introduction, e.g. what are the limitations and innovations?

Response: We gratefully appreciate for reviewer's valuable suggestions. As we emphasized in the introduction, designing innovative rapid modulation of intrinsic parallel synapses and applying it to build cross-layer modules helps achieve high-precision DNNs and direct cross-layer transmission of analog signals. However, the introduction should briefly state the purpose, scope, related work in the field, and theoretical foundations of the research, without extensively discussing the specific limitations of certain literature and basic circuit theories. Of course, as mentioned in the second paragraph of section 2 of the previous manuscript, we had discussed the existing works ("In previous reports...failing to propose effective strategies to leverage the dual-output potential of devices and achieve powerful network."), which is relatively brief. Readers without a solid understanding of neural network hardware may not fully comprehend the limitations of these works and the novelty of our proposed approach. Therefore, in this section, we will explain the limitations of all-electronic hardware neural networks and photonic hardware neural networks, and discuss the innovation of our work.

In previous all-electronic hardware neural network implementations, DAC/ADC and memory are essential for cross-layer signal transmission. This is because high-frequency modulation and parasitics severely degrades the quality of analog signal transmission, as mentioned in [*Nat Electron* 6, 680–693 (2023)], "All links with a Manhattan distance of 1 or 2 cores show no errors when run at 100 MHz, and 98% of them at 400 MHz. Links with longer Manhattan distances show more errors potentially due to attenuation from longer-distance routing metal wires due to

parasitics." Additionally, since the weight distribution in each layer is different, the resistivity varies, and all-electronic hardware neural networks typically use current as the computed result for each layer. According to Ohm's law, to ensure equipotential between two nodes, the current through high-resistance and low-resistance circuits is different. Therefore, it is almost impossible to directly divide a single computing current into multiple equal currents entering different layers. In all-electronic systems, ADCs are commonly used to convert the computed synaptic currents into digital signals for storage, and DACs are used to read the digital signals and convert them into corresponding voltage signals in layers that require access to these computed results [*Nature* 577, 641–646 (2020)]. This process consumes additional computational resources, and the throughput of digital signals is not as high as that of analog signals, thus limiting the further speed improvement of hardware neural networks.

In contrast, photonic neural networks can effectively avoid high-frequency interference and signal parallelization issues, as discussed in [*Nature* 588, 39–47 (2020)]. This is because optical signals themselves are immune to electrical signal interference, and the propagation of optical beams between layers does not interfere with each other. Therefore, previous reports have achieved end-to-end classification times of 570 ps, equivalent to a single clock cycle of state-of-the-art digital platforms [*Nature* 606, 501–506 (2022)]. However, photonic neural networks are difficult to be compatible with CMOS technology, dense fiber layout occupies a large area, and the cost of changing weights is extremely high once the hardware network is constructed, as weights are typically expressed using transmittance (fixed and unchangeable).

By summarizing the advantages and limitations of other works, our work innovatively utilizes light-emitting memristors as intrinsic parallel synaptic devices, effectively combining the synaptic tunability and integration advantages of hardware electronic neural networks with the interference immunity and parallelization advantages of photonic neural networks. Through optimization of the light-emitting memristors, we propose N-LEM with a UV negative differential conductance effect, effectively addressing the weight resetting issue of light-emitting memristors. We first construct a cross-layer transmission module based on N-LEM, achieving direct optoelectronic signal parallel transmission without the need for ADC/DAC conversion in computing results. Based on this hardware structure, we propose two ultra-deep neural networks with transfer learning capabilities through collaborative design of software and hardware. The proposed networks demonstrate performance that is close to or reaches the existing software level, providing a feasible and efficient solution for optoelectronic hybrid hardware neural networks.

The above is our detailed answer to the question. Of course, such a lengthy content is not suitable for inclusion in the manuscript. Therefore, we have modified the second paragraph of section 2, integrating the above answer into it after simplification. The specific content is as follows:

Page 5

"In previous all-electronic hardware neural network implementations, the use of DAC/ADC and memory is necessary for cross-layer signal transmission due to the negative effects of high-frequency modulation and parasitics on signal quality. All-electronic systems typically use ADCs to convert computed synaptic currents into digital signals for storage, and DACs are used to read these digital signals and convert them into corresponding voltage signals in layers requiring access to computed results. This process consumes additional computational resources and limits

the speed improvement of hardware neural networks as digital signals have lower throughput than analog signals. On the other hand, photonic neural networks effectively address the issues of high-frequency interference and signal parallelization. Optical signals are immune to electrical signal interference, and the propagation of light beams between layers will not cross talk with each other. This has enabled previous reports to achieve end-to-end classification times equivalent to a single clock cycle of state-of-the-art digital platforms. However, photonic neural networks face challenges in being compatible with CMOS technology, and the dense fiber layout occupies a large area. Furthermore, once the hardware network is constructed, changing weights becomes extremely costly as the weights are typically expressed using transmittance, which is fixed and unchangeable.”

Page 6

“Hence, a strategy utilizing dual-output characteristics of N-LEM to construct cross-layer transmission block (CIBlock) was proposed to achieve an ultra-deep photoelectric neural network (UPENN) and ultra-deep super-resolution reconstruction neural network (USRNN) with transfer learning ability (**Figure. 1b**), effectively combining the synaptic tunability and integration advantages of hardware electronic neural networks with the interference immunity and parallelization advantages of photonic neural networks.”

We hope that our explanations and amendments will be affirmed by the reviewers, and thank you again for your valuable advice.

2. The current-voltage characteristics in the N-LEM were investigated only in the positive voltage range. It is advised to investigate also in the negative voltage range.

Response: Thank you very much for your valuable suggestions. N-LEM, obtained by modifying the typical QLED structure to become a light-emitting memristor, must operate within a positive voltage range in order to function properly and produce light output [*Nano Lett.* 2021, 21, 6087–6094, *Nat Commun* 14, 2648 (2023)]. Additionally, if a negative voltage exceeding 3V is applied to the N-LEM, there is a high possibility of device breakdown, rendering it unusable. In fact, memristors operating in neural networks do not require operation under negative voltages because conductivity does not have negative values. Although synaptic weights in neural networks can be negative, the current mainstream approach is to use the Kirchhoff's law of two memristors to achieve the positive and negative expression of weights (As reported in [*Science* 381,1205-1211(2023)], the two-transistor-two-resistor configuration, namely two memristors, are used to represent a positive or negative weight.). Of course, as a QLED with a PIN structure, the N-LEM can operate at lower reverse voltages; however, in this case, it no longer functions as a memristor but serves as a photodetector instead. Nevertheless, this characteristic of the device under reverse voltage is irrelevant to the research content and innovation of this manuscript. The authors believe that adding the device's characteristics under reverse voltage in the manuscript may hinder reader comprehension. We provide this explanation and apologize for not modifying the content.

3. The voltage induced synaptic characteristics were analyzed as hole trapping in PVP layer. The authors need to explain this.

Response: Thank you very much for your valuable suggestions. A detailed explanation of how this hole trapping process leads to the manifestation of synaptic characteristics in the devices has been

provided in our previously reported work [*Nat Commun* 14, 2648 (2023)]. We mentioned this point in the manuscript and supplemented it with a mechanism diagram (**Figure. S3**) and corresponding process explanation in the SI:

In manuscript:

Page 7

“This process has been previously reported in our work¹³, and we have presented a schematic diagram and provided a detailed explanation of this process in **Figure. S3**”

In SI:

Supplementary Figure. S3 The working mechanism of PVP charge trapping layer. Part of the holes can be captured by the PVP layer when bias is applied (I). Then, when the bias is removed, the holes are stored in the PVP layer (II). When bias is applied again, trapped holes are released under the action of an applied electric field (III), thus increasing the conductivity and brightness of the device and achieving a simulation of synaptic plasticity.

“Polymer poly (4-vinyl phenol) (PVP) is a dielectric material, the polar groups contained in the PVP side group contain enormous amount of deep traps that allow charging and discharging carriers upon applied voltage. Therefore, in the previous report, this is the main reason for the hysteresis of OFETs with PVP as a dielectric layer [*Appl. Phys. Lett.* 89, 262120 (2006); *Appl. Phys. Lett.* 93, 143302 (2008); *Appl. Phys. Lett.* 108, 173301 (2016)]. Here, we exploit this feature to achieve the characteristics of artificial synapses by embedding capture layer PVP in QLED. Part of the holes can be captured by the PVP layer when bias is applied (I). Then, when the bias is removed, the holes are stored in the PVP layer (II). When bias is applied again, trapped holes are released under the action of an applied electric field (III), thus increasing the conductivity and brightness of the device and achieving a simulation of synaptic plasticity.”

4. As analyzed by the authors, after 2.52 s of UV irradiation of 50 pulse stimulation with an intensity of 70 mW/cm², the weight of the device is reduced by 100%. The authors should check the repeatability of the resetting time. They should show the consistency of resetting time with some number of setting and resetting operation. They also should show the resetting time with UV intensity.

Response: Thanks for reviewer’s suggestion. We have added the repeatability of the reset time for N-LEM, as shown in **Figure. S17**. In addition, we measured the reset time of the device current at 70mW/cm² UV irradiation after 10-50 pulses and the proportion of the device current drop under different optical power with fixed irradiation time, as shown in **Figure S18** and **S19**, respectively.

The corresponding descriptions have also been added to the manuscript.

In manuscript:

Page 7

“Furthermore, in **Figures. S17 to S19**, we demonstrate the repeatability of this reset operation, the reset time with different pulse settings, and the reset effect under different optical powers within a fixed duration.”

In SI:

Supplementary Figure. S17 Multi-cycle repetition of device reset operation after 50 pulse electric pulses.

Supplementary Figure. S18 Lighting time required for resetting with different pulse numbers (at 70 mW/cm²).

Supplementary Figure. S19 Weight reduction data under different optical powers after 50 electric pulses.

5. In supplementary Table-1, the UV negative photoconductive characteristics of various structures are indicated. As mention in the text, only the device with the IDBT/PVP/QDs(ZnO) exhibited significant negative UV photoconductivity. The authors need to put the exact photoconductivity value for different structures.

Response: Thank you very much for your valuable suggestions. We carefully reviewed supplementary Table-1 and found that one device structure was duplicated. The duplicate entry has been removed. Apart from the devices used in our main text, there are a total of 9 different control groups with distinct structures. We have provided additional data on the current changes of these 9 devices during the process of 50 electrical pulses, followed by a period of 3 seconds with or without UV light exposure, and subsequent continuous electrical pulse stimulation, as shown in **Figure. S8**. From the figure, it is evident that only the devices containing the IDTBT/PVP/QD(ZnO) structure exhibited a significant reduction in current, exceeding 20% compared to natural decay in the dark. Particularly, the structures containing IDTBT/PVP/QD/ZnO and IDTBT/PVP/QD showed an additional reduction in decay by over 91.47% and 68.17% respectively compared to the dark condition. This suggests that the devices with IDTBT/PVP/QD(ZnO) structure can effectively suppress electrical conductivity through the application of UV light.

We have revised the contents of relevant parts of the manuscript, supplemented **Figure. S8** and corrected **Supplementary Table 1** in SI, as follows:

In manuscript:

Page 9

“To investigate the negative UV photoconductivity mechanism of the device, 10 devices are fabricated with different structures (**Supplementary Table 1**). **Figure. S8** displays the current changes of 9 control structures during the process of 50 electrical pulse modulations with or without UV light exposure for 3 seconds, followed by continuous electrical pulse stimulation, electrical pulses and UV light are applied to all devices under identical conditions.”

In SI:

Supplementary Figure. S8 The current variation graphs of the 9 devices. By applying 50 continuous 5 V electrical pulses with an interval of 50 ms, followed by a 3 s exposure to both dark and ultraviolet light conditions, then the electrical pulse stimulation was administered again.

Supplementary Table 1

Table of UV negative photoconductive characteristics of various structures.

Number	Structure	UV negative photoconductivity (Yes/No)
0	ITO/PEDOT/PVK:IDTBT/PVP/QDs/ZnO/Ag	yes
1	ITO/PEDOT/IDTBT/PVP/QDs/ZnO/Ag	yes
2	ITO/PEDOT/PVK/PVP/QDs/ZnO/Ag	no
3	ITO/PEDOT/IDTBT/QDs/ZnO/Ag	no
4	ITO/PEDOT/IDTBT/PVP/ZnO/Ag	yes
5	ITO/PEDOT/IDTBT/PVP/QDs/Ag	yes
6	ITO/PEDOT/IDTBT/PVP/Ag	no
7	ITO/PEDOT/PVK/PVP/Ag	no
8	ITO/PEDOT/PVP/QDs/Ag	no
9	ITO/PEDOT/PVP/ZnO/Ag	no

6. The Fig. 3(c, d and f, g) are not clear. Are these figures only show electron distribution or hole distribution? If so, what is the meaning of positive and negative value according to the scale-bar. Also, it is recommended to keep same scale-bar with unit for all the figures.

Response: We are very grateful to the reviewer for your careful examination and pointing out our shortcomings. Fig. 3(c, d, f, g) is presented to show the concentration distribution of electrons and holes. Annotations (electron concentration, hole concentration) have been added above the modified four images. In addition, we attempted to standardize the scale bar for the four figures. However, since the distribution images and scale bars were obtained through software simulations, we did not find a way to standardize the scale bars. Therefore, we chose to add an annotation (*) below Figure. 3 to indicate the meaning of the scale bar. These scale bars were obtained by taking the logarithm with a base of 10 of the carrier quantity and are expressed in units of /m³. The modified **Figure. 3** is shown below:

Figure. 3 Mechanism and simulation of negative UV photoconductance effect.

(a) Formation and saturation process of built-in electric field in devices caused by UV light. (b) Device model and charge distribution in initial state. (c) Software simulation diagram of electronic distribution of devices in the dark. (d) Software

simulation diagram of electronic distribution of devices in the dark. (e) Device model, charge distribution and built-in electric field direction under UV irradiation. (f) Software simulation diagram of electron distribution and electric field direction of the device under UV light. (g) Software simulation diagram of holes distribution and electric field direction of the device under UV light.

7. Fig. S5, 10, 12, 13. show actual experimental data of the devices. The authors should not use ‘schematic diagram...’ in the caption.

Response: We are very grateful to the reviewer for your careful examination, and corrected these mistakes.

8. In many places there is no gap between numeric value and unit. The authors need to correct in whole manuscript.

Response: We are very grateful to the reviewer for your careful examination. We have carefully examined the whole manuscript and SI, and corrected these mistakes in the new version.

9. In the insets of Fig. 5(b and c) there is no label on x-axis.

Response: We are very grateful to the reviewer for your careful examination. The x-axis in **Figures. 5b** and **c** are different layers of neural networks in UPENN, more precisely, different layers of stage 1 and stage 4. We modified **Figures. 5b** and **c** and added the name of X-axis as position. In addition, we marked the positions of these network layers in **Figures. S21** and **S23** for readers' understanding. The modified **Figures. 5b, c** and content are as follows:

Figures. 5 (b) and (c) Gradient distribution calculated by selecting specific layers of stage 1 and stage 4 in UPENN and Contrast Net.

Page 14

“The gradient changes from deep to shallow in the first convolution layer of the CIBlocks is emphasized in stage 1 and stage 4 of both networks. (Figure. S21 and Figure. S23).”

10. It is recommended to use less abbreviations. Especially for a none expert reader, it is difficult to follow e.g. CLBlock, UPENN, USRNN. If possible, please avoid. Also, while I appreciate the extensive supplementary material, the reference to the supplementary material is too frequent. Maybe, the authors find a way to limit the reference to the supplementary material.

Response: Thank you very much for your valuable suggestions. We have considered that using too many new terms can be difficult for readers to understand. For example, the cross-layer block

(CIBlock), ultra-deep photoelectric neural network (UPENN), and ultra-deep super-resolution reconstruction neural network (USRNN) were mentioned by the reviewers. The purpose of using these abbreviations is to avoid making the language too long when referring to these structures and networks, which could impede readers' comprehension. We would like to minimize the use of abbreviations as it is more reader-friendly for non-specialized readers. However, our work innovatively proposes a strategy of directly constructing cross-layer structures through intrinsic parallel synapses, and develops two types of ultra-deep neural networks based on this strategy. These contributions are achieved through hardware-software co-design, involving novel hardware and network structures. To help readers better understand and grasp our proposed strategies and network structures, the use of abbreviations is essential.

We hypothesize that the reviewer's suggestion may be due to the fact that the full names of a large number of abbreviations are at the beginning of the article, which may require readers to repeatedly refer back to the beginning when reading the middle and later parts of the article, leading to cognitive jumps and reading barriers. Therefore, we chose to display the full names again when the above abbreviations were first used in each section of the article, providing explanations and emphasis to facilitate readers' understanding of our work.

Of course, to enhance understanding for non-specialized readers, we have removed references to BTNK in the manuscript, as BTNK is actually part of the CIBlock. This deletion helps prevent confusion or misunderstanding for readers, and in the Supplementary Information, we provide a detailed description of the relationship between the two (**Figure. S20**) in the section discussing the UPENN network structure. Furthermore, we have concentrated most of the abbreviation explanations at the second paragraph in section 2. This concentrated description clarifies the relationships among various abbreviations used in this work, allowing readers to establish the overall framework of our research from the beginning and better comprehend its content.

Furthermore, the reviewers mentioned our frequent references to supplementary materials, which are essential to provide a more comprehensive understanding of the principles of the new devices and network structures. However, we acknowledge that this frequent referencing was particularly noticeable in the original version of **Figure 5d, e, f** and **Figures. S19 to S22**, and that the original content was not intuitive or reader-friendly enough. Therefore, we made modifications to this section by revising the descriptions and removing the original Supplementary **Figures. S19 to S22**. Instead, we added another X-ray image for comparison to describe the process of feature disappearance. Detailed explanations and descriptions of this process are now provided below the new **Figures. S24 and S25**, enabling interested readers to better comprehend this process.

For this problem, we have added corresponding supplements in the article and SI, as follows:

In manuscript:

Page 6

“Hence, a strategy utilizing dual-output characteristics of N-LEM to construct cross-layer transmission block (CIBlock) was proposed to achieve an ultra-deep photoelectric neural network (UPENN) and ultra-deep super-resolution reconstruction neural network (USRNN) with transfer learning ability (**Figure. 1b**), effectively combining the synaptic tunability and integration advantages of hardware electronic neural networks with the interference immunity and parallelization advantages of photonic neural networks.”

Page 14

“The above results demonstrate that the cross-layer block (CIBlock) constructed by N-LEM is reliable. Furthermore, the existing Res-net model has been modified as the ultra-deep photoelectric neural network (UPENN) based on the cross-layer transmission module”

Page 15

“Different from shallow networks that can only perform single task, the ultra-deep photoelectric neural network (UPENN) is equipped with substantial neurons and synapses”

Page 18

“Here, we propose an ultra-deep super-resolution reconstruction neural network (USRNN) constructed using cross-layer block (CIBlock)”

In SI:

“Stage 1 to Stage 4 are different structures of CIBlock. CIBlock consists of BTNK 1 and BTNK 2, where BTNK 1 is the transmitting end of the optical interlayer signal, and BTNK 2 is the receiving end of the optical interlayer signal.”

“**Figure. S24** shows the feature maps obtained from stage 1 to stage 4 in UPENN for the X-ray images labeled as negative and positive in a binary classification task. In an ideal scenario, the neural network should extract completely different features for negative and positive cases in order to achieve close to 100% accuracy in distinguishing all images into two categories. It is evident from the figure that the two images acquire different features under the influence of the same convolutional kernels (the convolutional kernels are consistent at each position of the feature map), particularly in stage 3 and stage 4. This indicates that UPENN can effectively extract features by successfully avoiding gradient vanishing through cross-layer propagation and obtaining effective convolutional kernels through pre-training. By using SSIM to analyze the similarity between the two feature maps in stage 4, the SSIM coefficient is only 0.052, indicating that the two maps are extremely dissimilar. Therefore, UPENN is capable of effectively distinguishing between negative and positive cases.”

“**Figure. S25** shows the feature maps obtained from stage 1 to stage 4 in Contrast Net for the X-ray images labeled as negative and positive in a binary classification task. In an ideal scenario, the neural network should extract completely different features for negative and positive cases to achieve close to 100% accuracy in distinguishing all images into two categories. It is obvious from the figure that the two images have obtained very similar features under the influence of the same convolution kernel. This indicates that the convolution kernel of Contrast Net cannot effectively distinguish the image features of the two cases, which is essentially due to the invalid convolution kernel obtained in the pre-learning process caused by the gradient vanishing. By using SSIM to analyze the similarity between the two characteristic graphs in Stage 4, the SSIM coefficient is 0.925, which indicates that the two graphs are very similar. Therefore, Contrast Net cannot effectively distinguish negative and positive cases.”

Supplementary Figure. S24 The feature maps of two X-ray images in UPENN and the SSIM of two feature maps in stage 4.

Supplementary Figure. S25 The feature maps of two X-ray images in Contrast Net and the SSIM of two feature maps in stage 4.

11. While Figure 1 (a) to (c) look visually appealing, the content of the figures is questionable. Especially, Figure 1(a) does not provide the learning ability using a cross regional hierarchical structure.

Response: Thank you very much for your valuable suggestions. We propose for the first time a strategy of using intrinsic parallel synaptic devices to construct a direct cross-layer transmission module. Through hardware-software co-design, we have successfully implemented two efficient ultra-deep neural networks, which is a highly complex and challenging task. To spark readers' interest, we believe that more aesthetically pleasing schematic diagrams are advantageous. Therefore, **Figure. 1** serves as a simplified illustration of our research inspiration, network structure, and device, rather than providing a complete depiction. Regarding the reviewer's

comment "Figure 1(a) does not provide the learning ability using a cross regional hierarchical structure," the title of **Figure. 1(a)**, "A schematic diagram of pre-learning method and cross-regional hierarchical structure in biological brain. After pre-learning, the brain can process the unlearned content," aims to convey the importance of the biological brain's cross-regional hierarchical structure in learning, processing, and decision-making processes. Although this figure does not explicitly depict the cross-regional hierarchical structure, extensive research on this topic has been conducted previously. For example, Xi Yang et al. highlighted the significance of cross-regional hierarchical structures in visual information processing [*Artif Intell Rev* 55, 5263–5311 (2022)]; Tianqi Wan et al. mentioned that the brain needs to capture information from different brain regions to achieve auditory-motor integration [*Annu. Int. Conf. IEEE Eng. Med. Biol. Soc.* 2019, 3849-3853 (2019)]; Katja Ceranik et al. discovered extensive cross-regional feedforward and feedback connections among neurons [*J. Neurosci.* 17, 5380-5394 (1997)]. These studies fully demonstrate the presence of cross-regional hierarchical structures in the biological brain and their importance in enabling powerful brain functions. However, it does not imply that our work can achieve the same level of functionality as the brain. Due to the limitations of current neuroscience development, we have not yet fully understood how neurons and synapses work. Artificial neural networks inspired by this knowledge can only partially emulate brain functions. Given this context, previous research has already extensively demonstrated the necessity of cross-regional transmission structures and their functionalities in artificial neural networks. Therefore, it is reasonable to propose new strategies to overcome the limitations of previous works and implement such structures, and our results also demonstrate the effectiveness of this strategy.

We hope that our response can receive the reviewer's affirmation. We have modified the description of **Figure. 1a** and quoted the literature for readers to understand what is the cross-regional hierarchical structure of biology. The modification is as follows:

Page 5

"**Figure. 1a** illustrates a schematic diagram of pre-learning method and cross-regional hierarchical structure in biological brain³⁷⁻⁴⁰. The artificial neural network realized by imitating this structure shows great ability after extensive basic learning (pre-learning) and achieves high accuracy in untrained tasks, which is very similar to human learning ability."

Ref:

37 Ceranik, K. et al. A novel type of GABAergic interneuron connecting the input and the output regions of the hippocampus. *J. Neurosci.* 17, 5380-5394 (1997).

38 Peng, H. et al. Morphological diversity of single neurons in molecularly defined cell types. *Nature* 598, 174-181 (2021).

39 Wang, T. et al. Effect of Temporal Lobe Epilepsy on Auditory-motor Integration for Vocal Pitch Regulation: Evidence from Brain Functional Network Analysis. *Annu. Int. Conf. IEEE Eng. Med. Biol. Soc.* 2019, 3849-3853 (2019).

40 Yang, X. et al. Brain-inspired models for visual object recognition: an overview. *Artif. Intell. Rev.* 55, 5263-5311 (2022).

REVIEWERS' COMMENTS

Reviewer #1 (Remarks to the Author):

The authors' response and manuscript revisions addressed my concerns about compatibility and advantages with other electronic synaptic devices. The additional control group provided by the authors clearly demonstrates the differences between neural networks with and without cross-layer transmission structures, indicating the importance of CIBlock in constructing ultra-deep neural networks. The authors also responded to my question and reviewer #3's concern about frequent abbreviation usage, making corresponding revisions for better reader understanding. They discussed the pros and cons of current all-electronic memristor hardware neural networks and all-optical neural networks, explaining the innovation and key problems addressed in their work. I am satisfied with the revised manuscript and believe this work should be published in Nature Communications.

Reviewer #3 (Remarks to the Author):

The authors revised the manuscript "Cross-layer Transmission Realized by Light-emitting Memristor for Constructing Ultra-deep Neural Network with Transfer Learning Ability" to my full satisfaction and addressed all my comments and suggestion. Clearly, the authors did a great job. As mentioned in my previous report, I find the content of the manuscript and significance of the work performed excellent. I recommend the manuscript for publication.